# Taxonomic Study of Three Novel *Paenibacillus* Species with Cold-Adapted Plant Growth-Promoting Capacities Isolated from Root of *Larix gmelinii*

**DOI:** 10.3390/microorganisms11010130

**Published:** 2023-01-04

**Authors:** Han Xue, Yan Tu, Tengfei Ma, Ning Jiang, Chungen Piao, Yong Li

**Affiliations:** Key Laboratory of Biodiversity Conservation of National Forestry and Grassland Administration, Ecology and Nature Conservation Institute, Chinese Academy of Forestry, Beijing 100091, China

**Keywords:** *Paenibacillus* sp., plant growth-promoting bacteria, psychrotolerant, taxonomy

## Abstract

Exploration of the novel species of the genus *Paenibacillus* with plant-growth promoting characteristics at the low-temperature environment is of great significance for the development of psychrotolerant biofertilizer in forestry and agriculture. During the course of isolation of root endophytes of *Larix gmelinii* in the island frozen soil, three strains designated as T3-5-0-4, N1-5-1-14 and N5-1-1-5 were isolated. The three strains showed plant growth-promoting properties at the low temperature, such as phosphate solubilization, indole-3-acetic acid biosynthesis and siderophore production. According to pairwise sequence analyses of the 16S rRNA genes, the three strains represent putatively novel taxa within the genus *Paenibacillus*. The strains have typical chemotaxonomic characteristics of the genus *Paenibacillus* by having meso-diaminopimelic acid as diagnostic diamino acid, anteiso-C_15:0_ as the predominant fatty acid and MK-7 as the predominant menaquinone. The polar lipid profiles of all strains contained diphosphatidylglycerol, phosphatidylglycerol, and phosphatidylethanolamine. The sizes of the genomes of the stains ranged from 5.66 to 9.07 Mb and the associated G+C contents ranged from 37.9% to 44.7%. Polyphasic taxonomic study including determination of genome relatedness indices revealed that the strains are representatives of three novel species in the genus *Paenibacillus*. Consequently, isolates T3-5-0-4, N1-5-1-14 and N5-1-1-5 are proposed as novel species for which the names of *Paenibacillus endoradicis* sp. nov. (CFCC15691^T^ = KCTC43441^T^), *Paenibacillus radicibacter* sp. nov, (CFCC15694^T^ = KCTC43442^T^) and *Paenibacillus radicis* sp. nov. (CFCC15710^T^ = KCTC43173^T^), respectively. Moreover, analysis for biosynthetic genes showed that the strains have potential for plant growth-promoting characteristics, plant rhizospheres colonization and low-temperature adaption, most of which are consistent with the results of the bioactivity test.

## 1. Introduction

Plant growth-promoting bacteria (PGPB) are a group of beneficial microorganisms that exist widely in the rhizosphere soil or the inner tissue of plants and are closely related to the metabolism of the host plant roots. With the decreasing of the usage of pesticides and chemical fertilizers in contemporary agriculture, PGPR are gaining increasing attention. PGPB have great potential for improving plant yield as natural biofertilizers and it can be used as a sustainable long-term solution for soil fertilization [1], resulting in a reduced negative impact on the ecosystem and a more sustainable end-product [2]. PGPB carry out plant growth-promoting activities through direct and indirect mechanisms [3]. Direct mechanisms include nitrogen fixation from the atmosphere; phosphorus solubilization; siderophores synthesize; and plant hormone (indole, cytokinins, and gibberellins) production [4] to improve biotic and abiotic stress responses of plants [5,6,7]. The indirect mechanism contains activation of plant defenses, competition for resources and antimicrobial substances secretion to limit rhizosphere invasion by pathogenic organisms [8,9]. It is anticipated that one of the future research trends of PGPB will be the search for the novel and unstudied microbial species [10,11]. In this regard, the endosphere of plants represents a unique niche, as all studied plants have been found to have endophytic bacteria [12]; however, only around 2% of all known plant species have been investigated [13]. In the Northern Hemisphere’s low-temperature climatic zone, vegetation cycles begin at the beginning of spring or even in late winter, when temperatures are low and frosts occur occasionally. It has detrimental effects on the growth and plant growth-promoting (PGP) activities of mesophilic bacteria used as biofertilizers. At low temperatures, most microbial activities are slowed down or inhibited [14]. Therefore, screening for cold-resistant PGPB is important. Many studies showed that some species in the genus *Paenibacillus* isolated from inner tissues of plants can promote the growth of host plants through direct or indirect mechanisms [15,16,17,18,19,20], but there were no reports on psychrotolerant PGP *Paenibacillus*.

The genus *Paenibacillus* was proposed by Ash et al. [21] by separating ‘group 3’ within *Bacillus* into *Paenibacillus* according to the 16S rRNA gene sequences. At the time of writing, the genus *Paenibacillus* was a complex bacterial group with more than 270 species and 5 subspecies with validly published names (www.bacterio.net/paenibacillus.html (accessed on 1 November 2022)). Members of the genus *Paenibacillus* are rod-shaped, aerobic or facultatively anaerobic, Gram-positive or Gram-variable bacteria, and the DNA G+C contents ranges from 39 to 54 mol% [22]. The predominant cellular fatty acid and major respiratory quinone are Anteiso-C_15:0_ and menaquinone-7 [23,24]. The species of *Paenibacillus* are biochemically and genomically heterogeneous and occupy a wide range of niches [25], such as insect larvae, clinical samples, soil, and rhizosphere [26,27,28,29].

The Nanweng River Nature Reserve is in a high-latitude island of frozen soil in Greater Khingan, China. The annual average and lowest temperatures are −3 °C and −48 °C, respectively. Larch is the representative vegetation in this area. To explore the novel species of the genus *Paenibacillus* with plant-growth promoting characteristics of root tissue in *Larix gmelinii*, three strains were identified as novel taxa in the genus *Paenibacillus*. Thorough biosynthetic genes analysis, combined with bioactivity tests performed in this study, indicate a high potential for plant growth promotion at low temperatures of the three strains. As well as the polyphasic taxonomic studies, these strains were revealed as novel species based on their taxonomic provenance, for which the names *P. endoradicis* sp. nov., *P. radicibacter* sp. nov., and *P. radicis* sp. nov. were proposed. Our studies indicated that the novel species of *Paenibacillus* isolated from the root of *Larix gmelinii* have potential application values as a psychrotolerant biofertilizer in forestry or agriculture.

## 2. Materials and Methods

### 2.1. Isolation and Maintenance of the Strains

A collection of bacteria was isolated from the roots of *Larix gmelinii* from the Greater Khingan, Heilongjiang Province, PR China (51°07′23″–51°07′47″ N, 125°08′34″–125°08′39″ E). Following surface sterilization with 75% ethanol for 3 min and 2.5% sodium hypochlorite for 3 min, root tissues were washed with sterile distilled water [30]. Successful surface sterilization was confirmed by spreading the last washing water on Trypticase Soy Agar (TSA, Difco, Franklin Lakes, NJ, USA) plates without colony growth. Using sterile distilled water and silica sand, root samples were crushed and homogenized in 2 mL of the solution and diluted serially in sterile distilled water. Plates with 1/4 trypticase soy agar (TSA, Difco) and 1/4 nutrient agar (NA, Difco) were coated with the diluted solution and then incubated at 16 °C for 15–30 days. The pure colonies were obtained after repeated plate streaking and stored in cryotubes with 20% (*w*/*v*) glycerol at −80 °C.

### 2.2. Cultural, Morphological, and Physiological Characterization

Unless stated otherwise, the strains were observed on TSA for their cultural, morphological, and physiological characteristics. Gram staining of strains T3-5-0-4, N1-5-1-14 and N5-1-1-5 was conducted with a Gram staining kit (Sigma-Aldrich, St. Louis, MO, USA). Cultures were conducted at 4, 10, 15, 20, 25, 30, 37, 42, and 50 °C to determine the optimal temperature range for growth. The optimal pH range for growth was carried out using trypticase soy broth (TSB, Difco), with pH adjustments ranging from 4.5 to 10.0 (at intervals of 0.5 units) [31]. The NaCl requirement and tolerance were investigated by NaCl-free TSB supplemented with 0–10% (*w*/*v*) NaCl at 1% intervals. The growth tests were performed on Reasoner’s 2A (R2A, Difco) agar, Marine Agar 2216 (MA, Difco), nutrient agar (NA, Difco), and TSA at 25 °C for 14 days. The morphology and motility of cells were observed after 48 h of incubation at 25 °C by transmission electron microscopy (JEM-1230; JEOL) as well as phase-contrast microscopy (HFX; Nikon, Tokyo, Japan). The activity of constitutive enzymes, utilization of carbon source, and other physiological properties were analyzed by the API ZYM and API 20NE strips (bioMérieux, Marcy l’Etoile, France), and BIOLOG GEN III MicroPlate system (BIOLOG, Hayward, CA, USA).

### 2.3. Chemotaxonomic Characterization

Biomass was obtained from cultures grown on TSA medium at 25 °C for 48 h. An extraction, separation, and identification of fatty acid methyl esters were performed as instructed by the microbial identification system (MIDS). The extraction and analysis of polar lipids were conducted with two-dimensional thin-layer chromatography (TLC) following the description of Kates [32] and Minnikin [33]. Chloroform/methanol/water (65:25:4, by volume) and chloroform/methanol/acetic acid/water (85:12:15:4, by volume) were used for the two different dimensions. To identify individual polar lipids, phosphomolybdic acid, ninhydrin, and dragendorff reagent were sprayed over the surface of the sample. According to Schleifer and Kandler [34], the diaminopimelic acid isomers were identified using TLC in whole cell hydrolysates of strains. As described by Collins [35], the extraction and determination of isoprenoid quinones was performed using high-performance liquid chromatography.

### 2.4. Determination of Growth-Promoting Characteristics

#### 2.4.1. Phosphate Solubilization

Pikovskaya’s agar [36] was used to evaluate the P-solubilizing capacity of the isolates. The plates were inoculated and incubated at 10 °C for 14 days in triplicate. The solubilization halo (the translucent area surrounding the colony) diameter of the culture was measured. Based on these measurements, the Solubilization Index (SI) was calculated as follows: halo diameter (mm)/colony diameter (mm) [37].

#### 2.4.2. Indolic Compound Synthesis

A colorimetric assay based on Salkowsky’s reagent was used to estimate indolic compounds [38]. With the LB broth supplied with 0.15% tryptophan (*w*/*v*), bacteria were cultured at 10 °C and 90 rpm for 14 days in the dark in triplicates. The 1.5 mL suspension liquid was then transferred to a tube and centrifuged for 5 min at 12,000× *g*. The supernatant (0.5 mL) was added to and mixed thoroughly with the same volume of Salkowski reagent (1 mL 0.5 M FeCl_3_ with 49 mL of 35% HClO_4_
*v*/*v*). The mixture was transferred to cuvettes and incubated for 0.5 h in the dark. The mixture absorbance at 540 nm was determined using a spectrophotometer, and a calibration curve was constructed using pure indol-3-acetic acid.

#### 2.4.3. Siderophore Detection

The siderophore production of all isolates was evaluated using the Chromeazurol S (CAS) assay [39]. According to the instruction of Schwyn and Neilands [40], a freshly prepared sterile CAS reagent was used in each test, and the CAS reagent and solid medium were mixed at a 1:9 ratio for the experiment. Following spot inoculation, bacteria were incubated at 10 °C in triplicate for up to 21 days. Siderophore production is indicated by orange/yellow zones surrounding bacterial colonies. The production index (PI) = halo diameter (mm)/colony diameter (mm), which is a measure of the quantity secreted.

### 2.5. Genome Sequencing and Analysis

A Bacteria Genomic DNA Isolation Kit (Shanghai MajorBio Technologies Co. Ltd., Shanghai, China) was used to isolate the genomes of strains T3-5-0-4, N1-5-1-14 and N5-1-1-5, and then the Illumina NovaSeq PE150 (Beijing Novogene Bioinformatics Technology Co. Ltd., Beijing, China) was applied for whole-genome sequencing of them. Following the quality control of filtering raw sequences with the company’s compiling pipeline, the clean data were assembled with SOAP de novo software (http://soap.genomics.org.cn/soapdenovo.html (accessed on 21 March 2022)) [41]. Protein-encoding, ribosomal RNA (rRNA) genes, and transfer RNA (tRNA) were predicted and annotated using GeneMarkS [42], rRNAmmer [43], and tRNAscan-SE [44], respectively. Based on the COG database (Clusters of Orthologous Groups), gene functions were predicted and evaluated [45].

### 2.6. Phylogenetic Analysis

A PCR reaction was performed using universal bacterial primers 8F and 1525R. Purification and sequencing of the PCR products was carried out by Shanghai Sangon (Shanghai, China) on ABI 3730XL (Applied Biosystems, Waltham, MA, USA) using the dideoxy chain termination method [46]. The 16S rRNA gene sequence similarity was calculated by using the EzTaxon-e server [47]. A phylogenetic tree of 16S rRNA genes was constructed using the MEGA11 software package based on the gene alignment with the ClustalW program. Kimura’s 2-parameter model [48] was used to determine the genetic distances and clustering, and the phylogenetic trees were constructed using the maximum-likelihood (ML) [49], neighbor-joining (NJ) [50] and minimum-evolution (ME) [51] methods with 1000 replicates of bootstrap values to ensure the robustness of the conclusions. To elucidate the three strains’ exact taxonomic positions, a phylogenomic analysis was conducted with type reference strains using the UBCG pipeline. UBCG trees were generated using the alignment of 92 concatenated core genes of the genome, by excluding alignment positions with gap characters exceeding 50%. The GTR+CAT models and RAxML tool [52] were used to construct the phylogenomic trees based on the final nucleotide alignments. Another phylogenomic tree based on the codon tree method was constructed using the PATRIC server (https://patricbrc.org/app/PhylogeneticTree (accessed on 7 April 2022) [53], which performed the selection of single-copy PATRIC PGFams and analysis of aligned proteins and coding DNA from the genomes’ single-copy genes using the program RAxML [54].

### 2.7. Comparative Genomic Analyses

The DNA–DNA hybridization (dDDH) values were calculated using Genome-to Genome Distance Calculation (GGDC) version 2.1 (http://ggdc.dsmz.de/distcalc2.php (accessed on 9 April 2022) [55]. Average nucleotide identity (ANI) scores were calculated with the OrthoANI algorithm using the ANI calculator (www.ezbiocloud.net/toos/ani (accessed on 9 April 2022) [56].

## 3. Results

### 3.1. Morphological and Physiology Characteristics

More than 400 isolates were identified from the roots of *Larix gmelinii*. Isolate T3-5-0-4 was recovered from 1/4 TSA plates and isolates N1-5-1-14 and N5-1-1-5 were obtained from 1/4 NA plates. Cells from strains T3-5-0-4, N1-5-1-14, and N5-1-1-5 were Gram-stain-positive, aerobic, endospore-forming and rods with sizes of 3.8–4.0 × 0.6–0.7 μm, 3.2–3.8 × 0.8–1.2 μm, and 3.0–3.7 × 0.6–0.7 μm, respectively (Appendix A). Colonies grown on TSA agar are circular, smooth, and cream-colored after incubation for 2 days at 25 °C. The detection of growth of the three isolates in the presence of 0–3% (*w*/*v*) NaCl (optimum,0–2%), 0–3% (*w*/*v*) NaCl (optimum, 0%–1%), and 0–2% (*w*/*v*) NaCl (optimum,0–1%), and in the pH range from pH 7.0 to 8.0 (optimum, pH 7.0), pH 6.0 to 11.0 (optimum, pH 7.0–8.0) and pH 6.0 to 9.0 (optimum, pH 6.0–8.0), respectively. The growth temperature of all the three isolates was in the range of 4–37 °C (optimum, 20–25 °C). The culture grew on TSA, R2A, NA, and MA agar. Enzyme activities and biochemical analysis indicated that all three isolates were positive for catalase, oxidase, alkaline phosphatase, esterase lipase (C8), esterase (C4), naphthol-AS-BI-phosphoamidase, *α*-Glucosidase, *β*-Galactosidase, *β*-Glucosidase and indole production, and hydrolysis of aesculin, starch, p-nitrophenyl-*β*-D-galactopyranoside, and D-mannitol. all three isolates were negative for cystine arylamidase; *β*-Fucosidase; nitrate reduction; and assimilation of L-arginine, gelatin, capric acid, adipate, citric acid, phenylacetate, gluconate, and malic acid. In the carbon source utilization test, all three strains were positive for hydrolysis of starch, dextrin, D-trehalose, D-maltose, D-turanose, D-cellobiose, gentiobiose, D-salicin, *α*-D-glucose, *β*-methyl-D-glucoside, and D-galactose and negative for N-acetyl-D-galactosamine, 3-methyl glucose, N-acetylneuraminic acid, D-arabitol, D-fucose, D-serine, D-aspartic acid, L-serine, L-alanine, glycyl-L-proline, L-aspartic acid, L-arginine, L-glutamic acid, L-galactonic acid, quinic acid, mucic acid, p-hydroxy-phenylacetic acid, D-saccharic acid, D-lactic acid ethyl ester, citric acid, *α*-ketoglutaric acid, *γ*-amino-butryric acid, *α*-hydroxy-butyric acid, L-butyric acid, *β*-hydroxy-D, formic acid, and propionic acid. Other biochemical and enzymatic characteristics of the new isolates are presented in Table 1 and Appendix A.

### 3.2. Chemotaxonomic Characteristics

The major fatty acid profiles of strains T3-5-0-4, N1-5-1-14, and N5-1-1-5 compared to those of the closest-related species in the genus *Paenibacillus* are listed in Table 2. The predominant fatty acid in all three isolates was anteiso-C_15:0_, which was similar to the reference strains.

However, they were distinguished clearly from the reference species based on differences of fatty acid composition and content. The polar lipid profiles of all three isolates are shown in Appendix A. As a result of peptidoglycan analysis, meso-diaminopimelic acid was identified as a type of diamino acid, commonly observed in *Paenibacillus* species [57,58]. Menaquinone was the predominant isoprenoid quinone, with seven isoprene units (MK-7), as in other *Paenibacillus* species [23].

### 3.3. Plant Growth-Promoting (PGP) Traits

All the isolates T3-5-0-4, N1-5-1-14, and N5-1-1-5 showed PGP properties at low temperatures (10 °C) (Table 3). The strains N1-5-1-14 and N5-1-1-5 were capable of solubilization of phosphate and production of indole-3-acetic acid (IAA) and siderophores, whereas isolate T3-5-0-4 only could produce IAA (Table 3). The isolate N5-1-1-5 showed the highest IAA concentration and strongest capacity for siderophores production and phosphorus solubilization, and thus has the potential to be used as a psychrotolerant biofertilizer in forestry or agriculture after further research is completed.

### 3.4. Phylogenetic Analysis

Phylogenetic analysis based on pairwise comparisons of the 16S rRNA gene sequences revealed that all three strains showed the highest similarity values with *Paenibacillus* species. Strain N1-5-1-14 shared the highest similarity (96.02%) with *P. doosanensis* CAU 1055^T^, N5-1-1-5 shared 96.75% similarity with *P. rigui* JCM 16352^T^, and T3-5-0-4 shared 95.85% similarity with *P. paeoniae* M4BSY-1^T^. 16S rRNA gene similarities below the threshold value (98.65%) [59] for bacterial species division indicated that the three strains were mostly novel species of the genus *Paenibacillus*. Based on phylogenetic analysis of the 16S rRNA genes (Figure 1), strains T3-5-0-4, N1-5-1-14, and N5-1-1-5 formed distinct lineages and were affiliated with three clades of the genus *Paenibacillus*. These results were also supported by the neighbor-joining and minimum-evolution algorithms (Appendix A). In addition, a codon tree was constructed using the PATRIC server with single-copy genes in the genomes of isolates and relative *Paenibacillus* species for whole-genome phylogenetic analysis (Figure 2). Among the 35 genomes, 258 single-copy genes were detected and used to construct the codon tree. Three large clades were identified in the codon tree for members of the genus *Paenibacillus*, most of which were supported by 100% bootstrap value. Furthermore, the novel isolates formed distinct lineages and strains N5-1-1-5^T^ and N1-5-1-14^T^ fell into the same clade. The UBCG pipeline inferred a similar topology using the RAxML tool as well. (Figure 3).

### 3.5. Genome Feature and Annotation

The genome sizes of strains T3-5-0-4, N1-5-1-14, and N5-1-1-5 were 5.7, 5.4, and 9.0 Mb, with 12, 40, and 82 contigs, respectively. The G + C contents of genomic DNA for strains T3-5-0-4, N1-5-1-14, and N5-1-1-5 were 37.9, 42.4, and 44.7%, lower than those of the *Paenibacillus* species (Table 1). Moreover, three strains, T3-5-0-4, N1-5-1-14, and N5-1-1-5, were annotated using the GeneMarkS server, with 5252, 5130, and 8000 potential protein-coding genes, respectively. A comparison of the genomic features of the three strains is presented in Table 4. As the result of genome annotation, strains T3-5-0-4^T^, N1-5-1-14^T^, and N5-1-1-5^T^ harbored numerous genes associated with colonization of the plant rhizosphere (cheA/B/C/D/R/W/X/Y, MCP and motA/B genes), low-temperature adaption (heat-shock proteins, cold-shock proteins, RNA helicases and GntR genes), phosphate solubilization and transport (pstA/B/C/S and phnA/B/C/D/E/G/H/I/J/K/L/M/P/W/X genes), indole-3-acetic acid biosynthesis (AldB/H, miaA/B and trpA/B/C/D/E/F) and siderophore production (ABC.FEV.A/P/S and fbpA/B/C genes).

The three isolates contained the cheA/B/C/D/R/W/X/Y genes, which encode two-component chemotaxis systems and accessory proteins, and the MCP gene, which encodes methyl-accepting chemotaxis proteins (Appendix A). Furthermore, all three isolates possessed flagella biosynthesis genes as well as motA/B genes, indicating both a strong chemotactic response and increased motility [60,61,62]. Through chemotaxis and motility, strains move and accumulate in plant roots and gradually form a biofilm structure after reaching the plant root surface. Therefore, motility and chemotaxis are prerequisites for bacterial colonization of plant rhizospheres. The three isolates contained genes for cold-adapted proteins, such as heat-shock proteins (Hsps), cold shock proteins (Csps), and RNA helicases (Appendix A). These genes play a significant role in the cold adaptation of microorganisms. Iron is an important cofactor and electron acceptor in many enzymes and proteins that are essential to life. Siderophores produced by PGPB can scavenge non-heme and heme iron from the rhizosphere, increasing the availability of iron for plant hosts. The three strains were found to contain genes involved in the biosynthesis of siderophores. ABC.FEV.A/P/S is involved in iron complex transport and fbpA/B/C is involved in iron transport (II/III) (Appendix A). As a major restricted nutrient in soils, most of the phosphate is trapped as phosphonate, which cannot be incorporated biologically without degradation. The phosphate transport genes pstA/B/C/S and the phosphate catabolism genes phnA/B/C/D/E/G/H/I/J/K/L/M/P/W/X were detected in the three strains of chromosomal DNA (Appendix A). Despite the presence of iron transport and phosphate transport/catabolism genes in the genome of T3-5-0-4, they did not exhibit corresponding capacities. Gene expression is believed to be a complex process controlled by a variety of signaling substances, nutritional status, and environmental factors also influence gene expression. It is believed that not all the genes presented in the genomes are expressed. With the profound influences on the growth and development of plants, Indole-3-acetic acid (IAA) is the most important auxin endogenously produced by plant-associated bacteria. The aldehyde dehydrogenase coding genes—AldB/H, detected in the three genomes, have the potential to convert the indole acetaldehyde into IAA. The miaA and miaB genes encoding tRNA dimethylallyltransferase and tRNA-2-methylthio-N (6)-dimethylallyladenosine synthase were also identified (Appendix A). In addition, the genomes of the three strains contained trpA/B/C/D/E/F gene clusters, which are involved in L-tryptophan production.

### 3.6. Comparative Genomic Analyses

By comparing closely related type-strains with the three strains, it was clear that the strains were not related at the species level, as the ANI values (range 67.42%–72.50%) were well below the 95%–96% threshold recommended for species delineation [56,63] (Table 5). In addition, the GGDC values (range 21.2%–22.6%) were far below the threshold for species delineation of 70%. Moreover, the ANI and GGDC values determined from the genomic sequences of the three isolates indicated that the three strains were distinct from each other and did not have a clonal origin. These data suggest that strains T3-5-0-4, N1-5-1-14, and N5-1-1-5 represented three novel taxa within the genus *Paenibacillus*.

### 3.7. Discription of Paenibacillus endoradicis sp. nov.

*Paenibacillus endoradicis* (en.do.ra’di.cis. Gr. pref. *endo-*, in, within; L. fem. n. *radix* (gen. *radicis*), a root; N.L. gen. fem. n. *endoradicis*, of the inside of a root).

The cells are motile, Gram-stain-positive, endospore-forming, aerobic and rods with sizes of 3.8–4.0 × 0.6–0.7 μm. The culture is able to grow on TSA, R2A, NA and MA agars. Colonies grown on TSA are circular, smooth, creamcolored and semitransparent. Growth occurs between 4–37 °C (optimum, 20–25 °C), pH values of 7.0–8.0 (optimum, pH 7.0). and 0–3% (*w*/*v*) NaCl (optimum, 0–2%). Hydrolysis of starch, aesculin, urease, p-nitrophenyl-*β*-D-galactopyranoside, L-arabinose, D-mannose, D-mannitol and maltose. Positive for D-glucose fermentation, catalase, oxidase, alkaline phosphatase, esterase (C4), esterase lipase (C8), leucine arylamidase, valine arylamidase, naphthol-AS-BI-phosphoamidase, *α*-galactosidase, *β*-galactosidase, *β*-glucuronidase, *α*-glucosidase, and *β*-glucosidase. The principle fatty acids are anteiso-C_15:0_, C_16:0_ and iso-C_15:0_. The polar lipids comprise diphosphatidylglycerol, phosphatidylglycerol, phosphatidylethanolamine, aminophosphoglycolipid, three unidentified glycolipid, two unidentified aminophospholipid, unidentified phospholipid, and unidentified lipid. The diamino acid present in the cell wall peptidoglycan is meso-diaminopimelic acid. The predominant quinone is seven isoprene units (MK-7). The DNA G+C content is 37.9 mol%. The type strain (T3-5-0-4^T^ = CFCC15691^T^ = KCTC43441^T^) was isolated from the root of *Larix gmelinii* from Greater Khingan, Heilongjiang Province, PR China.

### 3.8. Discription of Paenibacillus radicibacter sp. nov.

*Paenibacillus radicibacter* (ra.di.ci.bac’ter. L. fem. n. *radix* (gen. *radicis*), root; N. L. masc. n. *bacter*, rod; N.L. masc. n. *radicibacter*, referring to a rod-shaped organism originating from a root).

The cells are motile, Gram-stain-positive, endospore-forming, aerobic, and rods with sizes of 3.2–3.8 × 0.8–1.2 μm. The culture is able to grow on TSA, R2A, NA, and MA agars. Colonies grown on TSA are circular, smooth, creamcolored, and semitransparent. Growth occurs between 4–37 °C (optimum, 20–25 °C), pH values of 6.0–11.0 (optimum, pH 7.0–8.0) and 0–3% (*w*/*v*) NaCl (optimum, 0–1%). Hydrolysis of starch, tweens 40, tweens 80, aesculin, p-nitrophenyl-*β*-D-galactopyranoside, D-glucose, D-mannitol, and N-acetyl-D-glucosamine. Positive for catalase, oxidase, alkaline phosphatase, esterase (C4), esterase lipase (C8), leucine arylamidase, valine arylamidase, trypsin, *α*-chymotrypsin, acid phosphatase, naphthol-AS-BI-phosphoamidase, *β*-galactosidase, *α*-glucosidase, *β*-glucosidase, and N-acetyl-*β*-glucosaminidase. The principle fatty acids are anteiso-C_15:0_, C_16:0_, iso-C_16:0_, and iso-C_15:0_. The polar lipids comprise diphosphatidylglycerol, phosphatidylethanolamine, aminoglycolipid, phosphatidylglycerol, unidentified aminolipid, four unidentified glycolipids, and three unidentified aminophospholipids. The diamino acid presenting in the cell wall peptidoglycan is meso-diaminopimelic acid. The predominant quinone is seven isoprene units (MK-7). The DNA G+C content is 42.4 mol%. The type strain (N1-5-1-14^T^ = CFCC15694^T^ = KCTC43442^T^) was isolated from the root of *Larix gmelinii* from Greater Khingan, Heilongjiang Province, PR China.

### 3.9. Discription of Paenibacillus radicis sp. nov.

*Paenibacillus radicis* (ra.di’cis. L. gen. fem. n. *radicis*, of a root, isolated from the root).

The cells are motile, Gram-stain-positive, endospore-forming, aerobic, and rods with sizes of 3.0–3.7 × 0.6–0.7 μm. The culture is able to grow on TSA, R2A, NA, and MA agars. Colonies grown on TSA are circular, smooth, creamcolored and semitransparent. Growth occurs between 4–37 °C (optimum, 20–25 °C), pH values of 6.0–9.0 (optimum, pH 6.0–8.0), and 0–2% (*w*/*v*) NaCl (optimum, 0–1%). Hydrolysis of starch, aesculin, p-nitrophenyl-*β*-D-galactopyranoside, L-arabinose, D-mannose, D-mannitol, and maltose. Positive for alkaline phosphatase, esterase (C4), esterase lipase (C8), lipase (C14), acid phosphatase, naphthol-AS-BI-phosphoamidase, *α*-galactosidase, *β*-galactosidase, *α*-glucosidase, *β*-glucosidase, N-acetyl-*β*-glucosaminidase, and *α*-mannosidase. The principle fatty acids are anteiso-C_15:0_, C_16:0_ and iso-C_14:0_. The polar lipids comprise phosphatidylmonomethylethanolamine, phosphatidylglycerol, diphosphatidylglycerol, phosphatidylethanolamine, phosphoglycolipid, two unidentified glycolipids, and six unidentified aminolipids. The diamino acid presenting in the cell wall peptidoglycan is meso-diaminopimelic acid. The predominant quinone is seven isoprene units (MK-7). The DNA G+C content is 44.7 mol%. The type strain (N5-1-1-5^T^ = CFCC15710^T^ = KCTC43173^T^) was isolated from the root of *Larix gmelinii* from Greater Khingan, Heilongjiang Province, PR China.

## 4. Discussion

In this study, three novel *Peanibacillus* species from the inner root of *Larix gmelinii* in the island frozen soil from the greater Khingan. The highest similarity values of 16S rRNA gene sequence between strains N1-5-1-14, N5-1-1-5, and T3-5-0-4 and members of the genus Peanibacillus showed 95.85–96.75%. Considering the species boundary of 98.65% for the 16S rRNA gene sequence similarity proposed by Kim, et al. [59], separate species positions of the isolated strains are indicated. The NJ, ML, and ME phylogenetic tree, based on 16S rDNA, showed that the isolates N1-5-1-14 and N5-1-1-5 formed two stable branches; meanwhile isolate T3-5-0-4 formed a cluster, which supported the three isolates was assigned to three novel species belonged to the genus *Peanibacillus*. It is known that phylogenetic analysis of 16S rRNA gene do not always provide the desired taxonomic resolution depending on the genus studied [64]. The similar topology was also obtained in the phylogenomic trees reconstructed by analyzing 258 single-copy coding sequences and 92 core genes of the genome sequences. Moreover, the genome similarity indices ANI and GGDC values between stains N1-5-1-14, N5-1-1-5, and T3-5-0-4 and Paenibacillus species were ranges 67.42%–72.50% and 21.2%–22.6%, respectively, being far below the recommended threshold for species demarcation [56,63], which provide more robust evidence to support the three strains do not have a clonal origin and represent three novel Paenibacillus speicies.

As for the physiological characteristics, hydrolysis of D-glucose fermentation and p-nitrophenyl-β-D-galactopyranoside and enzyme activities of alkaline phosphatase, esterase (C4), valine arylamidase, β-glucuronidase, α-glucosidase, and β-glucosidase could distinguish the three isolates from their closest relative species of *Peanibacillus* clearly. Moreover, the presence of APGL and GL in strain N1-5-1-14, AGL, GL, and AL in strain N5-1-1-5 and PME, PGL, and GL in strain T3-5-0-4 also distinguished them from the related Paenibacillus species. The predominant fatty acid and respiratory quinone of the strains were in accordance with those found in members of the genus *Paenibacillus* [23,24]. Based on the physiological and chemotaxonomic data, together with phylogenetic analyses, the three strains could be classified as three novel species in the *Paenibacillus* genus.

PGPB can promote plant growth through a variety of mechanisms, including biological nitrogen fixation, phosphate solubilization, 1-aminocyclopropane-1-carboxylate (ACC) deaminase activity, and production of siderophores and indolic compounds [65]. In this study, several plant-growth promoting activities, the solubilization of phosphate, and the production of indole-3-acetic acid (IAA) and siderophores have been detected in strains N1-5-1-14, N5-1-1-5, and T3-5-0-4 in a low temperature.

Phosphorus (P) is the second important plant growth-limiting nutrient, because the majority of soil P is found in insoluble forms and available phosphorous amount to plants is generally low [66]. In this context, the phosphate solubilizing microorganisms may provide the available forms of P to the plants. The inorganic and organic phosphorus were dissolved by low molecular weight organic acids and phosphatases synthesized by various soil bacteria [67,68]. As shown in Appendix A, the genome of strains N1-5-1-14, N5-1-1-5, and T3-5-0-4 contained PHN gene cluster (phnA/B/C/D/E/G/H/I/J/K/L/M/P/W/X), which is responsible for degrading phosphonates by bacteria, releasing bioavailable phosphate for plants nearby. Additionally, the phosphatases, responsible for Organic phosphate solubilization, was also detected in the genome of strain N1-5-1-14.

Iron exists mostly as Fe^3+^ in an aerobic environment and is likely to generate insoluble hydroxides and oxyhydroxides [69]. Typically, bacteria acquire iron by the release of iron chelators known as siderophores, with high association constants for complexing iron, which also have direct positive effects on plant development. The genes ABC.FEV.A/P/S involved in iron complex transport and fbpA/B/C involved in iron transport (II/III) were both detected in the three isolated strains. Compared with isolates T3-5-0-4, N1-5-1-14, and N5-1-1-5 had one more metabolic pathway, porphyrin, and chlorophyll metabolism, indicating that they may have a greater capacity for iron transport [70,71,72]. This was consistent with the siderophore production test results for these isolates.

All the three isolates genomes contained several genes related to IAA productions. For instance, aldehyde dehydrogenase encoding genes (AldB/H) indicated the strains could convert of indole acetaldehyde into IAA, and tRNA dimethylallyltransferase and tRNA-2-methylthio-N (6)-dimethylallyladenosine synthase-encoding genes (miaA/B), which are involved in the biosynthesis of creatine kinases N6-(dimethylallyl) adenosine (iPR) (by miaA) and 2-methylthio-N6-(dimethylallyl) adenosine (2MeSiPR) (by miaB) bound to tRNA. IAA not only plays a very important role in rhizobacteria–plant interactions [73] but also was implicated every aspect of growth and developmentof plants, as well as defense responses. Liu [74] reported 22 *Peanibacillus* strains could produce IAA at 30 °C with the IAA yield of 4.0–7.19 μg/mL. In present study, the three isolates produced the IAA of 5.3–9.0 μg/mL. It is clearly that the IAA productions of the three isolates were at a higher level, especially at a low temperature (10 °C).

Moreover, the three isolates contained cold-adapted proteins encoding genes, which played a significant role in the cold adaptation of microorganisms. As chaperones, Csps can regulate the secondary structure of mRNA and protein expression levels. Hsp70s maintain the correct translation process and assist in the assembly of intracellular molecules and folding of new proteins. GntR, a transcriptional regulator, is also involved in the regulation of cold responses. It is believed that these genes are important for the stains keeping biological activities at a low temperature, especially the PGP properties. In light of the above research results, we supposed that the three isolated strains were novel species of *Paenibacillus* with potential application values as a psychrotolerant biofertilizer in forestry or agriculture.

## 5. Conclusions

Based on the phenotypic and chemotaxonomic data, together with phylogenetic and genomic analyses, the three strains described represent three novel species in the genus *Paenibacillus* with potential applications as psychrotolerant biofertilizers in forestry and agriculture. For these, we propose the following names: *Paenibacillus endoradicis* sp. nov. for T3-5-0-4^T^, *Paenibacillus radicibacter* sp. nov. for N1-5-1-14^T^, and *Paenibacillus radicis* sp. nov. for N5-1-1-5^T^.

## Figures and Tables

**Figure 1 microorganisms-11-00130-f001:**
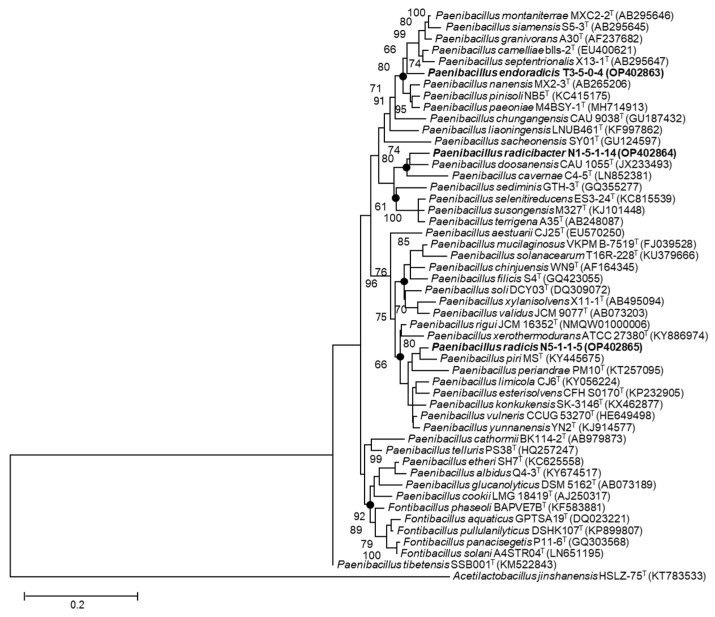
Maximum likelihood tree illustrating the phylogenetic position of T3-5-0-4, N1-5-1-14 and N5-1-1-5 and other type species in the genus *Paenibacillus* based on 16S rRNA gene sequences. The sequence of *Acetilactobacillus jinshanensis* HSLZ-75^T^ was used as the out-group. Closed circles indicate branches that were recovered with all three tree-making methods (maximum-likelihood, maximum-parsimony, and neighbor-joining). Bootstrap values (expressed as percentages of 1000 replications) over 60% are shown at branching nodes. Bar, 0.2 substitutions per nucleotide position.

**Figure 2 microorganisms-11-00130-f002:**
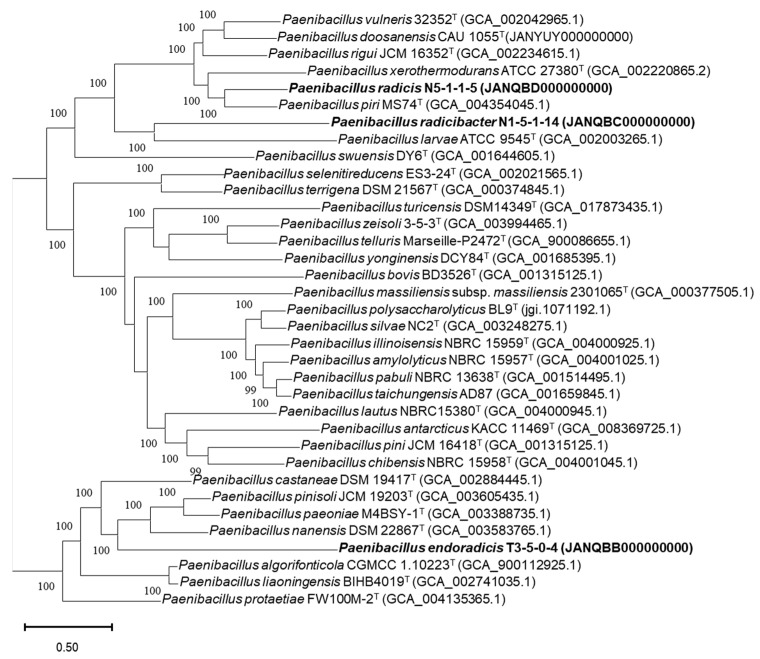
Phylogenetic tree based on whole genome comparison of of strains T3-5-0-4, N1-5-1-14, N5-1-1-5 and representatives of some related type strains in the genus *Paenibacillus* using RAxML within the codon tree pipeline at PATRIC, analyzing 258 single-copy coding sequences. Numbers at nodes represent confidence values of 100 rounds of fast bootstrapping. Bar, number of substitutions per site.

**Figure 3 microorganisms-11-00130-f003:**
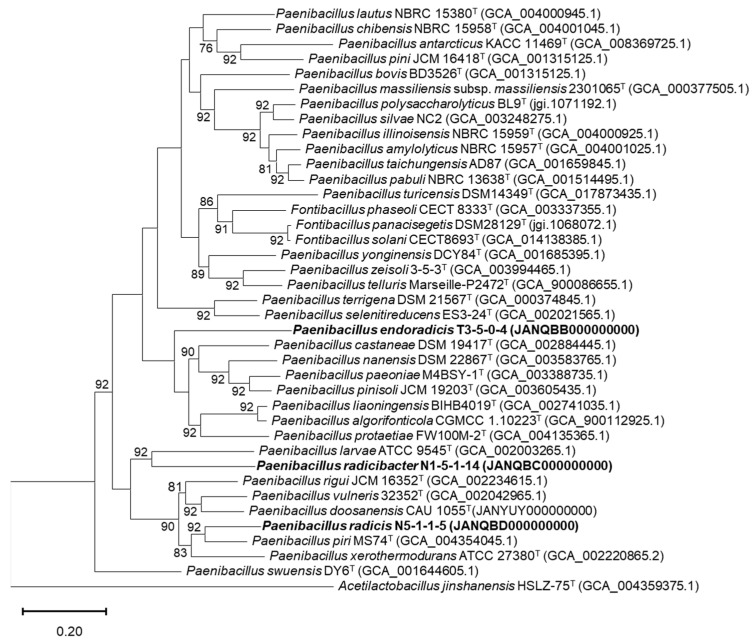
Phylogenetic tree based on the whole genome sequences showing the position of strains T3-5-0-4, N1-5-1-14, N5-1-1-5 and representatives of some related type strians in the genus *Paenibacillus*. The tree was constructed on concatenated alignment of 92 core genes (*alaS*, *argS*, *aspS*, *cgtA*, *coaE*, *cysS*, *dnaA*, *dnaG*, *dnaX*, *engA*, *ffh*, *fmt*, *frr*, *ftsY*, *gmk*, *hisS*, *ileS*, *infB*, *infC*, *ksgA*, *lepA*, *leuS*, *ligA*, *nusA*, *nusG*, *pgk*, *pheS*, *pheT*, *prfA*, *pyrG*, *recA*, *rbfA*, *rnc*, *rplA*, *rplB*, *rplC*, *rplD*, *rplE*, *rplF*, *rplI*, *rplJ*, *rplK*, *rplL*, *rplM*, *rplN*, *rplO*, *rplP*, *rplQ*, *rplR*, *rplS*, *rplT*, *rplU*, *rplV*, *rplW*, *rplX*, *rpmA*, *rpmC*, *rpmI*, *rpoA*, *rpoB*, *rpoC*, *rpsB*, *rpsC*, *rpsD*, *rpsE*, *rpsF*, *rpsG*, *rpsH*, *rpsI*, *rpsJ*, *rpsK*, *rpsL*, *rpsM*, *rpsO*, *rpsP*, *rpsQ*, *rpsR*, *rpsS*, *rpsT*, *secA*, *secG*, *secY*, *serS*, *smpB*, *tig*, *tilS*, *truB*, *tsaD*, *tsf*, *uvrB*, *ybeY*, and *ychF*). Bootstrap values are shown at branching nodes. Bar, 0.2 substitutions per nucleotide position.

**Table 1 microorganisms-11-00130-t001:** Differential characteristics of novel strains and type strains of related species of the genus *Paenibacillus*.

Characteristic	1	2	3	4	5	6
Colony color	cream	white	cream	cream	cream	cream
Ranges for growth						
NaCl tolerance (%, *w*/*v*)	0–3	0–5	0–3	0–4	0–2	0–1
Growth temperature (°C)	4–37	10–37	4–37	4–45	4–37	15–37
Growth pH	7.0–8.0	7.0–13.0	6.0–11.0	4.5–7.5	6.0–9.0	5.0–8.0
**Hydrolysis of**						
Starch	+	+	+	−	+	+
Tweens 40	−	−	+	−	−	−
Tweens 80	−	−	+	+	−	−
Nitrate reduction	−	−	−	−	−	+
D-glucose fermentation	+	−	−	+	−	+
Urease	+	−	−	−	−	−
*p*-nitrophenyl-*β*-D-galactopyranoside	+	−	+	−	+	−
D-glucose	−	+	+	+	−	+
L-arabinose	+	−	−	−	+	+
D-mannose	w	−	−	−	w	+
D-mannitol	+	+	+	−	+	+
N-acetyl-D-glucosamine	−	+	+	−	−	−
Maltose	+	+	−	+	+	+
Gluconate	−	−	−	−	−	+
Malic acid	−	−	−	−	−	+
**Enzyme activities**						
Alkaline phosphatase	+	−	+	+	+	−
Esterase (C4)	+	−	+	w	w	w
Esterase lipase (C8)	+	−	+	+	+	w
Lipase (C14)	−	−	−	−	+	−
Leucine arylamidase	+	+	+	−	−	+
Valine arylamidase	+	−	+	−	−	−
Trypsin	−	−	+	−	−	−
*α*-Chymotrypsin	−	−	+	+	−	−
Acid phosphatase	−	−	+	+	+	w
Naphthol-AS-BI-phosphoamidase	+	+	+	+	+	−
*α*-Galactosidase	+	w	−	+	+	+
*β*-Galactosidase	+	−	+	+	+	+
*β*-Glucuronidase	+	−	−	−	−	+
*α*-Glucosidase	+	−	+	+	+	−
*β*-Glucosidase	+	−	+	+	+	−
N-acetyl-*β*-glucosaminidase	−	−	+	−	+	−
*α*-Mannosidase	−	−	−	w	+	−
DNA G+C (%)	37.9	48.8	42.4	48.3	44.7	50.3

Taxa: 1, Paenibacillus endoradicis T3-5-0-4; 2, Paenibacillus paeoniae M4BSY-1^T^; 3, Paenibacillus radicibacter N1-5-1-14; 4, Paenibacillus doosanensis CAU 1055^T^; 5, Paenibacillus radicis N5-1-1-5; 6, Paenibacillus rigui WPCB173^T^. +, Positive; −, Negative; w, Weakly positive. All strains were motile, positive for catalase and oxidase, the hydrolysis of aesculin, and indole production and negative for cystine arylamidase, β-fucosidase, the hydrolysis of L-arginine, gelatin, capric acid, adipate, citric acid, and phenylacetate.

**Table 2 microorganisms-11-00130-t002:** Cellular fatty acid compositions (%) of novel strains and the type of strains of the most closely related *Paenibacillus* species.

	1	2	3	4	5	6
Saturated						
C_12:0_	/	1.1	/	/	/	/
C_14:0_	1.5	1.9	/	/	3	/
C_15:0_	/	5.7	/	/	/	5.6
C_16:0_	23.1	16.2	9.3	3.8	10.8	3.1
C_17:0_	/	2.1	/	/	/	1.4
C_18:0_	/	/	/	/	1.8	/
Unsaturated						
C_16:1_ω11c	/	2.5	/	2.5	/	/
C_17:1_ω9c	2.6	1.5	2.5	/	3.1	2.9
iso-C_17:1_ω10c	/	/	/	1.1	/	/
C_18:1_ω9c	/	3.7	/	/	/	/
Branched-chain						
iso-C_14:0_	1.2	2.8	3.8	2.3	5.5	3.2
iso-C_15:0_	5.1	5.7	8.0	7.5	3.5	7.5
anteiso-C_15:0_	56.1	25.0	58.5	53.7	63.9	63.5
iso-C_16:0_	1.5	12.8	8.7	15.9	4.9	5.7
iso-C_17:0_	/	5.9	1.3	3.2	/	/
anteiso-C_17:0_	3.9	8.6	3.6	5.6	1.8	1.9

Strains: 1, Paenibacillus endoradicis sp. nov. T3-5-0-4; 2, Paenibacillus paeoniae M4BSY-1^T^; 3, Paenibacillus radicibacter sp. nov N1-5-1-14; 4, Paenibacillus doosanensis CAU 1055^T^; 5, Paenibacillus radicis sp. nov. N5-1-1-5; 6, Paenibacillus rigui WPCB173^T^. All data from this study. Only those fatty acids amounting to >1.0% in all strains are shown.

**Table 3 microorganisms-11-00130-t003:** Plant growth promoting properties of novel strains. SI, solubilization Index; IAA, indole-3-acetic acid; PI, production index.

Strain Number	Plant Growth Promoting Attributes		
	SI	IAA (μg/mL)	PI
T3-5-0-4	-	6.2 ± 0.7	-
N1-5-1-14	1.2 ± 0.1	5.3 ± 0.5	1.1 ± 0.1
N5-1-1-5	1.5 ± 0.2	9.0 ± 0.8	1.7 ± 0.3

**Table 4 microorganisms-11-00130-t004:** General genome characteristics of (1) *Paenibacillus endoradicis* T3-5-0-4, (2) *Paenibacillus radicibacter* N1-5-1-14, and (3) *Paenibacillus radicis* N5-1-1-5.

Characteristics	1	2	3
Assembled contigs	12	40	82
Genome size (Mb)	5.66	5.42	9.07
NO. contigs	12	40	82
N50	859,719	337,399	238,693
Largest contig size	1,355,421	992,081	629,400
Protein-coding genes (CDS)	5252	5130	8000
rRNAs (5s, 16s, 23s)	2,0,0	8,1,1	11,1,0
tRNAs	51	83	87

**Table 5 microorganisms-11-00130-t005:** Genomic similarities (%) of ANI and GGDC Values between the three isolated strains and other relatives type species in the genus *Paenibacillus*. (1) *Paenibacillus endoradicis* T3-5-0-4, (2) *Paenibacillus radicibacter* N1-5-1-14, (3) *Paenibacillus radicis* N5-1-1-5, (4) *Paenibacillus paeoniae* M4BSY-1^T^, (5) *Paenibacillus pinisoli* NB5^T^, (6) *Paenibacillus doosanensis* CAU 1055^T^, (7) *Paenibacillus turicensis* MOL722^T^, (8) *Paenibacillus rigui* JCM 16352^T^, and (9) *Paenibacillus vulneris* CCUG 53270^T^.

Reference Strains	ANI/GGDC Value (%)		
	1	2	3
1	100	/	/
2	67.92/21.30	100	/
3	68.35/22.60	68.45/21.20	100
4	69.28/21.50	67.57/22.30	68.43/21.80
5	69.19/20.30	67.64/23.50	68.03/22.20
6	68.14/25.90	68.02/21.60	71.96/19.10
7	68.21/19.20	67.77/21.10	67.78/24.40
8	67.70/24.50	68.25/21.00	72.50/19.40
9	67.42/25.30	68.48/20.10	72.00/18.90

## Data Availability

The 16S rRNA gene sequence of *Paenibacillus endoradicis* T3-5-0-4, *Paenibacillus radicibacter* N1-5-1-14, and *Paenibacillus radicis* N5-1-1-5 have been deposited in the GenBank database with the accession numbers OP402863, OP402864, and OP402865, respectively. The GenBank accession numbers for the draft genome sequence of *Paenibacillus endoradicis* T3-5-0-4, *Paenibacillus radicibacter* N1-5-1-14, and *Paenibacillus radicis* N5-1-1-5 are JANQBB000000000, JANQBC000000000, and JANQBD000000000, respectively.

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
