# Peer review of "Taxonomic Study of Three Novel Paenibacillus Species with Cold-Adapted Plant Growth-Promoting Capacities Isolated from Root of Larix gmelinii"

_microorganisms, 2023, doi:10.3390/microorganisms11010130_

Round 1
Reviewer 1 Report
The supplementary data file (tables and figures is not included
Introduction is short, needs enrichment
M&M
Add reference for the isolation and surface sterilization procedure
Line 70: What do you mean by 1/4 trypticase soy agar, 1/4 nutrient agar?
In a growth tests , what is the need for incubation for 14 days?
On what basis did you choose these three isolates?
You have been mentioned in Materials and Methods that the NaCl tolerance test was conducted using concentrations of 1-10% salt, but what was mentioned in the results is that the salt concentration used is 0-3% ?
Section 2.3 Chemotaxonomic characterization; you wrote that you will identify :
- fatty acid methyl esters
- polar lipids
- diaminopimelic acid isomers
- isoprenoid quinones
while the results in table 2 listed the cellular fatty acid compositions only?
The author reported that, the three isolates were negative for indole production (line 182, 198), Conversely, the results recorded in Table 3 and in lines 323, 324 indicate the ability of the three isolates to produce indole?
Please write what the abbreviations (SI and PI) in Table 3 refer to?
You stated that " Despite the presence of iron transport and phosphate transport/catabolism genes in the genome of T3-5-0-4T, they did not exhibit corresponding capacities", Do you have an explanation for that?
The entire discussion is devoid of comparative studies and references.
Replace Figure 1 with a clearer one
Line 14; "aceticacid" insert space "acetic acid"
Line 43: delete "by"
Lines 195-197: write bacterial names in italic
Line 196: correct "Paenibacillus radices" to "Paenibacillus radicis"
Line 223: correct "Table 2" to "Table 3"
Line 384: write " Larix gmelinii" in italic
In Table 2, write the title above the table, the table key below the table. As in Table 1
In Table 5: write the synonyms for numbers 1, 2 and 3 in the headers of columns no 2, 3 and 4

Author Response
Response to Reviewer 1 Comments
We appreciate reviewer #1 for his/her effort to review our manuscript, and his/her positive feedback. The reviewer gives an accurate summary of our work and brings forward constructive questions. We have addressed them below. Changes highlighted in yellow have been made accordingly in the revised manuscript.
Point 1: Your manuscript seemed interesting to me, but I have doubts about the correct choice of the journal section (Systems Microbiology) for submitting your manuscript, as well as the choice of Microorganisms. It is not clear what the purpose and hypothesis of the study was. If the the goal of the manuscript was to validate three new species of the genus Paenibacillu, then the part of the manuscript devoted to polyphasic taxonomic studies (16S rRNA, fatty acids, quinones, genome relatedness index, etc.) and the description of new species of microorganisms should have been submitted to the International Journal of Systematic and Evolutionary Microbiology. A new species is validated only after being published in the IJSEM Notification List. If the purpose of the study was to describe the genetic features and PGP properties of three new strains of the genus Paenibacillus, then the title of the manuscript should be changed, and its taxonomic description sections of the new species should be deleted.
Response 1: Thank you for your revision suggestion and it is of great help to improve the quality of the manuscript. The purpose of the study is to explore the novel species of the genus Paenibacillus with plant-growth promoting characteristic of root tuissues in Larix gmelinii. I did some work before submitting to Microorganisms and noticed that many articles about systematic studies of novel bacterial taxon with special functions has been published on the this journal, such as “Stieleria sedimenti sp. nov., a Novel Member of the Family Pirellulaceae with Antimicrobial Activity Isolated in Portugal from Brackish Sediments” ( https://doi.org/10.3390/microorganisms10112151), “Genomic and Phylogenetic Characterization of Rhodopseudomonas infernalis sp. nov., Isolated from the Hell Creek Watershed (Nebraska)” ( https://doi.org/10.3390/microorganisms10102024), and “Genomic Analysis and Characterization of Pseudotabrizicola formosa sp. nov., a Novel Aerobic Anoxygenic Phototrophic Bacterium, Isolated from Sayram Lake Water” (https://doi.org/10.3390/microorganisms10112154). Our manuscript described three new species of Paenibacillus, and these three strains have shown growth-promoting characteristic at low temperatures. Therefore, I think the manuscript is in line with submission requirements of Microorganisms.
Point 2: In Abstract (line 12) strains NC2-4-308T, NC3-4-326T and NA3-4-109T are erroneously indicated.
Response 2: The strains “NC2-4-308T, NC3-4-326T and NA3-4-109T” has been changed to “T3-5-0-4, N1-5-1-14 and N5-1-1-5”.
Point 3: The Introduction should clearly state the purpose and hypothesis of the study.
Response 3: The purpose of this study has been state in the introduction.
Point 4: Line 50-51. Provide a link to sources that describe psychrotolerant PGP Paenibacillus.
Response 4: I’m sorry for this mistake. I haven’t found the report about psychrotolerant PGP Paenibacillus,so I amended the sentence to “Many studies showed that some species in the genus Paenibacillus isolated from inner tu-issues of plants can promote the growth of host plants through direct or indirect mecha-nisms [15-20], but there were no reports on psychrotolerant PGP Paenibacillus.”
Point 5: I believe that it is wrong to designate the strains studied in the manuscript as type strains (NC2-4-308, not NC2-4-308T). In addition, in the dendrograms (Fig. 1-3) the studied strains, in my opinion, are better denoted as NC2-4-308, NC3-4-326 and NA3-4-109.
Response 5: The number of three isolates are T3-5-0-4, N1-5-1-14 and N5-1-1-5, and all the “T” have been deleted all through the manuscript.
Point 6: In sections 2.1 and 2.2, the composition of the culture media should be added.
Response 6: The mediums were bought from the company Difco, and the “Difco” has been added after the culture media
Point 7: Section 3 Results and discussion should be divided into separate sections (3. Results, 4. Discussion). At the same time, the Discussion section should be expanded.
Response 7: The Section 3 has been divided into two separate sections—Results and Discussion. The Discussion section has been expanded.
Point 8: There are no Supplementary Materials (figures and tables) in my personal reviewer profile, which makes reviewing very difficult.
Response 8: The Supplementary Materials have been uploaded to the system and I don’t know why the they could not be viewed. I will upload them again this time.
Point 9: Lines 217-218, 283-285, 301-303. References to literature should be provided.
Response 9: The references 57-59 have been added to Lines 217-218, and references 61-63 have been added to Line 283-285. The lines 301-303 have been moved to the Discussion section, and the references 73-75 has been added.
Point 10: Lines 277-780. Examples of specific genes (several for each property) associated with colonization of the plant rhizosphere, low-temperature adaptation, solubilization and transport of phosphates, production of IAA and siderophores should be given.
Response 10: The specific genes associated has been added.
Point 11: In the note to Table 3, I recommend that you indicate the decoding of the designations SI, PI, IAA. Also, when discussing the results, it is necessary to compare the level of IAA production with other PGP strains Paenibacillus.
Response 11: The decoding of SI, PI, IAA have been noted before Table 3. The IAA production with other PGP strains Paenibacillus have been added in the discussion.

Reviewer 2 Report
Dear Аuthors!
Your manuscript seemed interesting to me, but I have doubts about the correct choice of the journal section (Systems Microbiology) for submitting your manuscript, as well as the choice of Microorganisms. It is not clear what the purpose and hypothesis of the study was. If the the goal of the manuscript was to validate three new species of the genus Paenibacillu, then the part of the manuscript devoted to polyphasic taxonomic studies (16S rRNA, fatty acids, quinones, genome relatedness index, etc.) and the description of new species of microorganisms should have been submitted to the International Journal of Systematic and Evolutionary Microbiology. A new species is validated only after being published in the IJSEM Notification List. If the purpose of the study was to describe the genetic features and PGP properties of three new strains of the genus Paenibacillus, then the title of the manuscript should be changed, and its taxonomic description sections of the new species should be deleted.
In addition, to improve the quality of the manuscript, I propose to make a number of corrections to it.
1) In Abstract (line 12) strains NC2-4-308T, NC3-4-326T and NA3-4-109T are erroneously indicated.
2) The Introduction should clearly state the purpose and hypothesis of the study.
3) Line 50-51. Provide a link to sources that describe psychrotolerant PGP Paenibacillus.
4) I believe that it is wrong to designate the strains studied in the manuscript as type strains (NC2-4-308, not NC2-4-308T). In addition, in the dendrograms (Fig. 1-3) the studied strains, in my opinion, are better denoted as NC2-4-308, NC3-4-326 and NA3-4-109.
5) In sections 2.1 and 2.2, the composition of the culture media should be added.
6) Section 3 Results and discussion should be divided into separate sections (3. Results, 4. Discussion). At the same time, the Discussion section should be expanded.
7) There are no Supplementary Materials (figures and tables) in my personal reviewer profile, which makes reviewing very difficult.
8) Lines 217-218, 283-285, 301-303. References to literature should be provided.
9) Lines 277-780. Examples of specific genes (several for each property) associated with colonization of the plant rhizosphere, low-temperature adaptation, solubilization and transport of phosphates, production of IAA and siderophores should be given.
10) In the note to Table 3, I recommend that you indicate the decoding of the designations SI, PI, IAA. Also, when discussing the results, it is necessary to compare the level of IAA production with other PGP strains Paenibacillus.
Author Response
Response to Reviewer 2 Comments
We appreciate reviewer #2 for his/her effort to review our manuscript, and his/her positive feedback. The reviewer gives an accurate summary of our work and brings forward constructive questions. We have addressed them below. Changes highlighted in yellow have been made accordingly in the revised manuscript.
Point 1: The supplementary data file (tables and figures is not included)
Response 1: The Supplementary Materials have been uploaded to the system and I don’t know why the they could not be viewed. I will upload them again this time.
Point 2: Introduction is short, needs enrichment
Response 2: The introduction has been enrichment.
Point 3: Add reference for the isolation and surface sterilization procedure
Response 3: The reference 30 has been added for the isolation and surface sterilization procedure.
Point 4: Line 70: What do you mean by 1/4 trypticase soy agar, 1/4 nutrient agar?
Response 4: The 1/4 trypticase soy agar or 1/4 nutrient agar means diluting the medium by a factor of 4.
Point 5: In a growth tests, what is the need for incubation for 14 days?
Response 5: The 14-day incubation period is enough for the strain to be verified to grow on different medium.
Point 6: On what basis did you choose these three isolates?
Response 6: Actually, we collected all strains with different morphologies on the plates, and extracted the genome DNA of these strains. The 16S rRNA genes were amplified and sequence similarity was calculated by using the EzTaxon-e server. According to the similarity of 16S rRNA genes, three novel strains of genus Peanibacillus were selected.
Point 7: You have been mentioned in Materials and Methods that the NaCl tolerance test was conducted using concentrations of 1-10% salt, but what was mentioned in the results is that the salt concentration used is 0-3%?
Response 7: The NaCl requirement and tolerance of the three isolates were investigated by NaCl-free TSB supplemented with 0- 10% (w/v) NaCl at 1% intervals. However, the growth of the three isolates were detected only in the presence of 0-3% (w/v) or 0-2% (w/v) NaCl.
Point 8: Section 2.3 Chemotaxonomic characterization; you wrote that you will identify:
fatty acid methyl esters
polar lipids
diaminopimelic acid isomers
isoprenoid quinones
while the results in table 2 listed the cellular fatty acid compositions only?
Response 8: The polar lipids profiles were shown in Fig. S2 and Table S7, which has been stated in the manuscript below the table 2. The diaminopimelic acid isomers and isoprenoid quinones were clarified in manuscript below the table 2.
Point 9: The author reported that, the three isolates were negative for indole production (line 182, 198), conversely, the results recorded in Table 3 and in lines 323, 324 indicate the ability of the three isolates to produce indole?
Response 9: I’m sorry for this mistake. The three isolates do produce indole and the content has been amended.
Point 10: Please write what the abbreviations (SI and PI) in Table 3 refer to?
Response 10: I have added the interpretation of abbreviations at the top of Table 3.
Point 11: You stated that " Despite the presence of iron transport and phosphate transport/catabolism genes in the genome of T3-5-0-4T, they did not exhibit corresponding capacities", Do you have an explanation for that?
Response 11: Gene expression is believed to be a complex process controlled by a variety of signaling substances, nutritional status, and environmental factors also influence gene expression. It is believed that not that all the genes presented in the genomes are expressed.
Point 12: The entire discussion is devoid of comparative studies and references.
Response 12: The part of result and discussion has been devided into two sections. The comparative studies and references have been added in the section Discussion.
Point 13: Replace Figure 1 with a clearer one
Response 13: The fig 1 has been replaced by a clearer one.
Point 14: Line 14; "aceticacid" insert space "acetic acid"
Response 14: The space has been inserted
Point 15: Line 43: delete "by"
Response 15: The “by” has been deleted.
Point 16: Lines 195-197: write bacterial names in italic
Response 16: The bacterial name has been rewritten in italic.
Point 17: Line 196: correct "Paenibacillus radices" to "Paenibacillus radicis"
Response 17: The words "Paenibacillus radices" have been corrected to "Paenibacillus radices".
Point 18: Line 223: correct "Table 2" to "Table 3".
Response 18: The "Table 2" has been corrected to "Table 3".
Point 19: Line 384: write " Larix gmelinii" in italic
Response 19: The words " Larix gmelinii" have been rewritten in italic
Point 20: In Table 2, write the title above the table, the table key below the table. As in Table 1
Response 20: The table key of Table 2 has been moved below.
Point 21: In Table 5: write the synonyms for numbers 1, 2 and 3 in the headers of columns no 2, 3 and 4
Response 21: I’m sorry that I could not got the means of this question clearly.

Round 2
Reviewer 2 Report
Dear Аuthors!
Thank you for considering my comments on the manuscript.